# Quatsomes Loaded with Squaraine Dye as an Effective Photosensitizer for Photodynamic Therapy

**DOI:** 10.3390/pharmaceutics15030902

**Published:** 2023-03-10

**Authors:** Nicolò Bordignon, Mariana Köber, Giorgia Chinigò, Carlotta Pontremoli, Ettore Sansone, Guillem Vargas-Nadal, Maria Jesus Moran Plata, Alessandra Fiorio Pla, Nadia Barbero, Judit Morla-Folch, Nora Ventosa

**Affiliations:** 1Institut de Ciència de Materials de Barcelona, ICMAB-CSIC, Campus UAB, Bellaterra, 08193 Catalonia, Spain; 2Department of Life Sciences and Systems Biology, University of Torino, Via Accademia Albertina 13, 10123 Turin, Italy; 3CIBER de Bioingeniería, Biomateriales y Nanomedicina, Instituto de Salud Carlos III, 08193 Bellaterra, Spain; 4Department of Chemistry, NIS Interdepartmental Centre and INSTM Reference Centre, University of Torino, Via Quarello 15a, 10135 Turin, Italy

**Keywords:** nanovesicles, quatsomes, squaraine, photodynamic therapy

## Abstract

Photodynamic therapy is a non-invasive therapeutic strategy that combines external light with a photosensitizer (PS) to destroy abnormal cells. Despite the great progress in the development of new photosensitizers with improved efficacy, the PS’s photosensitivity, high hydrophobicity, and tumor target avidity still represent the main challenges. Herein, newly synthesized brominated squaraine, exhibiting intense absorption in the red/near-infrared region, has been successfully incorporated into Quatsome (QS) nanovesicles at different loadings. The formulations under study have been characterized and interrogated in vitro for cytotoxicity, cellular uptake, and PDT efficiency in a breast cancer cell line. The nanoencapsulation of brominated squaraine into QS overcomes the non-water solubility limitation of the brominated squaraine without compromising its ability to generate ROS rapidly. In addition, PDT effectiveness is maximized due to the highly localized PS loadings in the QS. This strategy allows using a therapeutic squaraine concentration that is 100 times lower than the concentration of free squaraine usually employed in PDT. Taken together, our results reveal the benefits of the incorporation of brominated squaraine into QS to optimize their photoactive properties and support their applicability as photosensitizer agents for PDT.

## 1. Introduction

Photodynamic Therapy (PDT) is a minimally invasive localized clinical treatment that has been developed to treat many diseases, including psoriasis [1,2] or several types of cancer [3,4,5]. PDT is based on the presence of three components: a photosensitizer (PS), light, and molecular oxygen. Studies on cells and animals started in the 1960s and led to the clinical approval by the Food and Drug Administration (FDA) of the first photosensitizer, Photophrin, in 1995 [5,6]. In PDT, PSs are exposed to light at a specific wavelength, depending on the nature of the molecule in use [7]. After irradiation, the PS absorbs the light, causing the electron to transition from its ground state (singlet state) to an excited singlet state. Then, the PS can lose energy and return to the ground state. Alternatively, the singlet state can undergo intersystem crossing (ISC), forming an excited triplet state caused by the spin conversion of the electron in the higher energy orbital. From this triplet state, the molecule can relax and go back to the singlet state via two different routes: (i) the molecule can reduce the substrate forming radicals, which then react with the oxygen, producing oxygenated radicals (Reactive Oxygen Species, ROS), known as a Type I reaction, or (ii) the PS can directly react with molecular oxygen, producing singlet oxygen (^1^O_2_), known as a Type II reaction [8,9]. These reactions also explain the importance of oxygen’s presence in PDT. Both products then induce apoptosis or necrosis, causing damage to tumor cells and tumor-associated vascular structures, contributing to the stimulation of the immune response in the host [5,10].

For a safe and effective photodynamic treatment, the PS must be non-cytotoxic in the dark but highly cytotoxic after irradiation, photo- and chemically stable, non-mutagenic, and selective against neoplastic tissues. In addition, it should present a high degree of purity and ideally absorb light between 600 and 800 nm to promote a deeper tissue penetration and minimize light scattering by tissues [11,12]. Most of the FDA-approved or currently under clinical trials PS [13,14] are based on porphyrins or chlorin structures, such as Photofrin [15] or Foscan [16,17], porfimer sodium and temoporfin, respectively. Those PS, based on an extended aromatic ring system, are highly hydrophobic and susceptible to π–π stacking. This results in poor solubility in aqueous media and rapid clearance in blood circulation, severely compromising the therapeutic effectiveness of PDT. In this context, organic dyes, such as polymethine dyes (squaraine and cyanine dyes), represent a promising alternative to PS thanks to the higher selectivity, purity, and absorption at longer wavelengths compared to porphyrin-derived PS [18,19,20,21,22]. Indeed, both cyanine and squaraine dyes have shown excellent light-induced toxicity on different types of tumors.

Despite the many merits of improving PS performances, the poor water solubility (leading to aggregation in aqueous media) and low chemical stability are still the main challenges for their biomedical application [21,23,24]. In order to improve the PSs’ performances and to protect them from photodegradation, different delivery systems have been developed [24,25,26,27], with liposomes and polymeric micelles being the most common ones. These colloidal nanostructures—formed by the self-assembly of amphiphilic molecules in water—are used for the encapsulation of non-water-soluble PS in either the hydrophobic core of micelles [28] or the membrane of liposomes [29,30]. It is worth noting that other strategies, such as the insertion of functional groups into the PS structure to increase water solubility, have been reported [31]. However, those modifications usually compromise the photochemical properties of the PS [32]. Thus, the nanoencapsulation of the PS improves its solubility in the aqueous environment without altering its chemical structure and benefits from either active targeting, with the functionalization of targeting agents [33,34,35], or passive targeting through the enhanced permeation and retention effect (EPR) [36]. However, most of those formulations are limited by the lack of stability over time. For example, liposomes tend to change morphology, aggregate, or suffer from PS leakage over time [37,38], thus requiring complex formulations and coatings [39] to overcome these aspects.

To address these challenges, in the present work, we have employed non-liposomal nanovesicles, named quatsomes (QS), as a nanocarrier for a hydrophobic squaraine dye. QS are non-liposomal thermodynamically stable nanometric vesicles with very low dispersity [40] composed of sterols and quaternary ammonium surfactants [41]. These sterols and surfactants self-assemble in water, forming amphiphilic spherical nanometric structures with high homogeneity. This kind of vesicle has been proven to be safe and non-toxic for biomedical applications and has remained stable for years [42,43]. The encapsulation of PSs into QS is an attractive approach since it offers not only a strategy to bring non-soluble squaraine dyes into an aqueous media but also a list of advantages for its in vivo application: (i) longer times in circulation and higher cellular uptake [44,45], (ii) improved therapeutic efficiency with a lower dye concentration since the photosensitizer is highly localized at the therapeutic side [42,46] and, (iii) QS allows a targeted delivery by the nanovesicle functionalization with targeting units [47,48].

In previous work, Bromo-Squaraine-C4 (Br-Sq-C4), a newly synthesized squaraine photosensitizer, demonstrated successful results in vitro [49]. However, a small organic molecule has important limitations for its in vivo application, such as low solubility and poor spectroscopic properties in aqueous media and a tendency to aggregate. Thus, as a first trial, non-water soluble Br-Sq-C4 was incorporated into QSs nanovesicles to enhance its stability in aqueous media. Nonetheless, Br-Sq-C4 entrapment efficiency was lower than 50% after preparation and not stably anchored into the QS membrane over time, showing significant dye leaking with only 15% of the initial dye concentration remaining after 6 weeks (see Appendix A). Considering the low dye loading efficiency as well as the instability of this system, the use of Br-Sq-C4-loaded QS as a potential photosensitizer agent was dismissed. Instead, we synthesized a similar squaraine bearing longer alkyl chains (**Br-Sq-C12**, see Figure 1), i.e., C12 instead of C4 hydrocarbon chains. By the incorporation of this longer hydrocarbon chain, **Br-Sq-C12** can be anchored to the QS membrane stably due to its larger lipophilicity.

Herein, in this work, we present the design of a new QS composed of Cholesterol and surfactant Sterealkonium Chloride loaded at different concentrations of **Br-Sq-C12** (Figure 1). First, we studied the physicochemical and spectroscopic characteristics of newly synthesized Br-Sq-C12-loaded quatsomes. The entrapment of Br-Sq-C12 into the QSs does not interfere with its ability to generate ROS rapidly, an essential requirement for PDT activity. Cellular uptake and PDT efficiency are then studied in vitro in a cancer cell model showing the benefits of loading Br-Sq-C12 into a QS vs. Br-Sq-C12 in its free form. In addition to overcoming the non-water solubility limitation of Br-Sq-C12, the nanometric volume facilitated by the QS allows highly localized PS loadings, maximizing in this way the PDT effectiveness. This Br-Sq-C12-loaded QS can not only be explored for the development of highly efficient PDT treatment against cancer but also offers a basis to attain the development of photosensitizers with improved characteristics for in vivo applications.

## 2. Materials and Methods

### 2.1. Synthesis of Bromo-Squaraine-C12 Dye

All the chemicals were purchased from Merck (Darmstadt, Germany), Alfa Aesar (Haverhill, MA, USA), or TCI (Tokyo, Japan) and were used without any further purification. All microwave reactions were performed in single-mode Biotage Initiator 2.5 (Biotage, Uppsala, Sweden). TLC was performed on silica gel 60 F254 plates. ^1^H NMR (600 MHz) spectra were recorded on a Bruker Avance 600 NMR (Bruker, Billerica, MA, USA) in CDCl_3_. ESI-MS spectra were recorded using an LTQ Orbitrap (Thermo Scientific, Waltham, MA, USA) spectrometer, with an electrospray interface and ion trap as a mass analyzer. The flow injection effluent was delivered into the ion source using nitrogen as sheath and auxiliary gas.

As previously reported, 5-bromo-2,3,3-trimethyl-3H-indole (**1**) was synthesized [50].

#### 2.1.1. Quaternarization Synthesis of Indolenine 5-Bromo-1-dodecyl-2,3,3-trimethyl-3H-indol-1-ium Iodide

All together, 5-bromo-2,3,3-trimethyl-3H-indole (**1**) (500 mg, 2.1 mmol), iodododecane (1.6 mL, 6.3 mmol), and anhydrous acetonitrile (10 mL) were introduced in a reaction vial, sealed with a crimp cap and heated in a microwave system at 155 °C for 60 min. At the end of the reaction, acetonitrile was removed in the rotavapor, and diethyl ether (200 mL) was then poured to precipitate a white-brownish solid, which was washed three times with diethyl ether and filtered (287 mg, 25.5% yield).

^1^H NMR (600 MHz, CDCl_3_): δ 7.78 (dd, J = 8.6, 1.8 Hz, 1H), 7.73 (d, J = 1.8 Hz, 1H), 7.57 (d, J = 8.6 Hz, 1H), 4.65 (t, 2H), 2.99 (s, 3H), 1.68 (s, 6H), 1.24 (m, 20H), 0.87 (t, J = 7.0 Hz, 3H).

^13^C NMR: The compound solubility proved too low to record a ^13^C NMR spectrum.

#### 2.1.2. Synthesis of Br-Sq-C12

Compound **2** (287 mg, 0.54 mmol), 4-dihydroxycyclobut-3-ene-1,2-dione (30.8 mg, 0.27 mmol), and 5 mL of a mixture of toluene and n-butanol (1:1) were introduced inside a sealed MW vial and heated up to 160 °C for 30 min. The solution turned blue, and the TLC (Pet. Ether: Acetone 8:2) showed reaction completion. After solvent evaporation, column chromatography (Pet. Ether: Acetone 70:30) afforded **Br-Sq-C12** as a golden solid (140 mg, yield = 60%). ^1^H NMR and ESI-MS spectra are shown in Appendix A, respectively.

^1^H NMR (600 MHz, CDCl_3_): δ 7.44 (s, 2H), 7.41 (d, J = 8.3 Hz, 2H), 6.83 (d, J = 8.3 Hz, 2H), 5.94 (s, 2H), 3.92 (s, 4H), 1.77 (s, 12H), 1.24 (m, 40H), 0.87 (t, J = 6.9 Hz, 6H).

^13^C NMR: The compound solubility proved too low to record a ^13^C NMR spectrum.

HRMS (ESI) *m*/*z*: [M]^+^ calcd for [C_50_H_71_Br_2_N_2_O_2_]^+^ 889.3877, found 891.3869.

### 2.2. Preparation of Dye-Loaded Chol/Stk QS by DELOS-Susp

All the QS formulations described were prepared using the DELOS-susp method [47,51]. The employed quantities (Appendix A) and the detailed protocol used are listed in the Appendix A. To prepare the organic phase of the DELOS-susp, the desired amount of Chol (PanReac AppliChem, Castellar del Vallès, Spain) and Stk (TokyoChemical Industry CO. LTD, Tokyo, Japan) was dissolved in ethanol (HPLC grade purity, Avantor Performance Materials Poland S.A., Silesia, Poland). This ethanolic solution contains the already solubilized **Br-Sq-C12** (see Table 1 for the exact concentration of each component). The solution was introduced in the high-pressure vessel, and the compressed CO_2_ was added, with a final temperature of 38 °C and 11.5 MPa of pressure. After one hour, the expanded solution with all the membrane components dissolved was depressurized over the desired amount of water. After the production, the dye-loaded Chol/Stk QS was purified using diafiltration to remove ethanol and the non-incorporated dye and membrane components.

#### Determination of the Dye Concentration and Dye Loading in QS Nanovesicles

The concentration of dye entrapped in QS was determined by measuring the UV–Vis absorbance *A* using a UV–Vis spectrophotometer (V-780, Jasco, Easton, Sweden) and a high precision cell (Hellma Analytics, Müllheim, Germany) with a pathlength *l* of 1 cm. All the samples were diluted in ethanol to disrupt the membrane and release all the entrapped dye molecules. The concentration *C* of **Br-Sq-C12** was determined using the Lambert–Beer law (A = ε l C), knowing that the extinction coefficient (ε) of **Br-Sq-C12** in EtOH is 290.484 M^−1^ cm^−1^.

The dye-loading coefficient was determined by lyophilization of the samples (LyoQuest-80, Telstar, Terrassa, Spain) at 193 K and 5 Pa for 5 days. Then, the samples were weighted, and the loading in mass was determined through the following equation.
$$\mathrm{dye}\;\mathrm{loading}=\frac{\mathrm{mass}\;\mathrm{of}\;\mathrm{dye}}{\mathrm{mass}\;\mathrm{of}\;\mathrm{all}\;\mathrm{vesicle}\;\mathrm{components}-\mathrm{mass}\;\mathrm{of}\;\mathrm{dye}}$$

### 2.3. Spectroscopic Characterization of Free Br-Sq-C12 and Dye-Loaded QSs

#### 2.3.1. UV–Vis Spectroscopy, Molar Extinction Coefficient, and Solvatochromism

To determine the molar extinction coefficient of the **Br-Sq-C12**, different dilutions in ethanol were prepared from a stock solution (0.5 mM). The absorbances were measured and their maxima were plotted vs. the sample concentration, being the slope of the linear fitting of the molar extinction coefficient (ε). The analysis was performed in duplicate and data were considered acceptable when the difference between the measured log ε was equal to or lower than 0.02 to their average.

The determination of the solvatochromism was performed by preparing different solutions of **Br-Sq-C12** and dye-loaded QSs in acetone, absolute ethanol (EtOH), methanol (MeOH), double distilled water (ddH_2_O), and Dimethyl sulfoxide (DMSO). The absorption was measured at room temperature by UV–Vis spectroscopy (Cary 300 Bio spectrophotometer, Varian, Santa Clara, USA or V-780, Jasco, Easton, Sweden) in the range of 500–800 nm using quartz cuvettes, using a 1 cm path length.

#### 2.3.2. Fluorescence Spectroscopy

Fluorescence emission measurements were acquired in steady-state mode and recorded in the range of 595–750 nm using a Horiba Jobin Yvon Fluorolog 3 TCSPC fluorimeter (Kyoto, Japan) equipped with a 450-W Xenon lamp and a Hamamatsu R928 photomultiplier (Hamamatsu photonics, Hamamatsu, Japan) by using solvents with different polarity to investigate the solvatochromic behavior of both the **Br-Sq-C12** and dye-loaded QS. The excitation wavelength was different depending on the solvents and was set at the squaraine hypochromic shoulder previously recorded at the UV–Vis spectra. The excitation and emission slits were 5 nm and 5 nm, respectively.

Fluorescence quantum yields (QY) were determined using the same instrument with Quanta-φ integrating sphere and De Mello method. The QY was evaluated in absolute ethanol for the **Br-Sq-C12** and ddH_2_O for the dye-loaded QSs. The analyzed samples had an absorbance of around 0.1 to avoid aggregations/fluorescence quenching. The final result is an average of three independent measurements of different dye solutions.

Fluorescence lifetimes (LT) were determined using the time-correlated single photon counting method (Horiba Jobin Yvon, Horiba, Kyoto, Japan) using a 636 nm Horiba Jobin Yvon NanoLED (Horiba, Kyoto, Japan) as the excitation source and a pulse repetition frequency of 1 MHz positioned at 90° with respect to a TBX-04 detector. Lifetimes were calculated using DAS6 decay analysis software. The LT was evaluated in absolute ethanol for the **Br-Sq-C12** and ddH_2_O for the dye-loaded QSs.

### 2.4. Physicochemical Characterization and Stability of Dye-Loaded QS

#### 2.4.1. Dynamic Light Scattering (DLS) and Electrophoretic Light Scattering (ELS)

The mean size and size distribution of the QS loaded with 200 and 300 μM **Br-Sq-C12** (QS_Sq_160 and QS_Sq_200, respectively) were determined by DLS, while the determination of the ζ-potential values (z-pot) was performed with the ELS. Both measurements were carried out using a Zetasizer Ultra (Malvern Instruments, Malvern, UK). The measurements with the DLS technique were performed using a fluorescence filter to block the light resulting from fluorescence emission, which may alter the correlation function (the instrument exploits a 633 nm laser). Meanwhile, the ELS measurements were performed employing A DTS1070 folded capillary cell (Malvern Instruments, Malvern, UK) was used, applying a voltage of 40 mV between the gold electrodes, and being calculated using the Helmholtz–Smoluchowski equation, which can potentially underestimate the real zeta-potential [52,53]. All the measurements were performed in triplicate to ensure the reliability of the results.

#### 2.4.2. Cryogenic Transmission Electron Microscopy

Cryogenic transmission electron microscopy (cryo-TEM) images were acquired with a JEOL JEM microscope (JEOL JEM 2011, Tokyo, Japan) operating at 200 kV under low-dose conditions. First, 10µL of the sample was deposited onto the holey carbon grid and, immediately after, vitrified by rapid immersion in liquid ethane. The vitrified sample was mounted on a cryo-transfer system and introduced into the microscope (Gatan 626, Gatan, Pleasanton, CA, USA). Images were recorded on a CCD camera (Gatan Ultrascan US1000, Gatan, Pleasanton, CA, USA).

### 2.5. Evaluation of ROS Generation with DPBF and DCFH

As a probe molecule, 1,3-Diphenylisobenzofuran (DPBF, Sigma Aldrich, Darmstadt, Germany) was used to evaluate Reactive Oxygen Species (ROS) generation by following the protocol previously described in the literature [49]. DPBF rapidly reacts with ^1^O_2_ forming the colorless *o*-dibenzoylbenzene derivative. The ^1^O_2_ scavenger activity can be monitored through a decrease in the electronic absorption band of DBPF at 415 nm. Stock solutions were prepared in DMSO, absolute ethanol, and phosphate buffer (2 mM, pH 7.4), respectively, for DPBF, free **Br-Sq-C12**, and **Br-Sq-C12**-loaded quatsomes. Each solution was then diluted in phosphate buffer (2 mM, pH 7.4) to obtain a DPBF concentration of 25 μM and a final concentration of 2.5 μM for both the free and the encapsulated dye. The solutions were placed in a 1 cm quartz cell and irradiated at various time intervals under stirring in an aerated solarbox (Solarbox 3000e, 250 W xenon lamp, CO.FO.ME.GRA, Milan, Italy). The light was filtered in an optical filter with a 515 nm cut-off to avoid DPBF degradation. At predefined time points (30, 60, 90, 120, and 180 s), absorption spectra were recorded on a Cary 300 Bio spectrophotometer instrument (Varian, Santa Clara, CA, USA). The decrease in the DBPF absorption contribution at 415 nm was plotted as a function of the irradiation time.

### 2.6. Biological Assays

#### 2.6.1. Cell Culture, Cell Viability, and Phototoxicity Assay

Human breast adenocarcinoma cell line (MCF-7, ECACC, European Collection of Authenticated Cell Cultures, Porton Down, UK) was cultured in DMEM High Glucose (Euroclone, Pero, Italy) growth medium complemented with 10% Fetal Bovine Serum (Euroclone, Pero, Italy), 100 mg/mL PenStrep (Sigma-Aldrich, Saint Louis, MO, USA), and 2 mM L-glutamine (Sigma-Aldrich, Saint Louis, MO, USA). Cells were cultured in a humidified incubator (HeraCell 150, Heraeus, Hanau, Germany) with 5% CO_2_ at 37 °C, using Falcon™ plates as supports.

To investigate QS’s cytotoxicity, MCF-7 cells (0.5·10^4^ cells/well) were seeded in 96-well plates (Sarstedt, Nümbrecht, Germany). Six hours after plating, cells were treated with QS_Blank at two different membrane components’ (Chol + Stk) final concentrations (10 μg/mL and 2 μg/mL). Cell viability was assessed using CellTiter 96^®^ Aqueous Non-Radioactive cell proliferation assay (Promega, Madison, WI, USA) according to the manufacturer’s instructions 24, 48, and 72 h after treatment. Briefly, 2 h after MTS incubation at 37 °C, absorbance at 490 nm was recorded using a microplate reader (FilterMax F5, Multi-Mode Microplate Reader, Molecular Devices, San Jose, MO, USA). Absorbance values were normalized on the control at 24 h and analyzed as being proportional to the number of viable cells. Similarly, the cytotoxicity of QS (2 μg/mL) loaded with increasing dye concentrations (Table 2) was assessed.

To evaluate the photodynamic effect of QS_Sq, MCF-7 cells (0.5·10^4^ cells/well) were seeded in 96-well plates. Six hours after plating, cells were treated with QS_Sq at the concentrations reported in Table 2 and **Br-Sq-C12** in its free form at the same concentrations. After O/N incubation at 37 °C and 5% CO_2_, the cells were irradiated for 15 min with a RED-LED array (96 LEDs in a 12 × 8 arrangement, excitation wavelength: 640 nm, and irradiance: 8 mW/cm^2^) specifically designed and produced by Cicci Research s.r.l (Grosseto, Italy). Cell viability was assessed 24, 48, and 72 h after irradiation using CellTiter 96^®^ AQueous Non-Radioactive cell proliferation assay (Promega, Madison, WI, USA) as described above. The photodynamic effect of **Br-Sq-C12**-loaded QS was evaluated by comparing the viability of cells treated with QS_Sq or with the same concentration of **Br-Sq-C12** in its free form upon irradiation. For each condition, eight technical replicates were set up and three independent experiments were performed.

#### 2.6.2. Cellular Uptake

To verify the intracellular uptake of Sq-loaded QS and compare it with that of the dye in its free form, Calcein (Molecular probes^®^, Invitrogen, Waltham, MA, USA) has been used to label and track the whole cellular volume in MCF-7 live cells. Briefly, MCF-7 cells were seeded in Ibidi μ-Slide 8 wells (1.6 × 10^4^ cells/well), and 24 h after seeding, cells were treated overnight with 85 nM of Br-Sq-C12 in its free form or incorporated within 2 μg/mL QS (QS_Sq_200). After the incubation with the QS_Sq_200, the cells were washed twice with PBS and incubated with Calcein (500 nM) for 30 min, washed twice with Hanks’ Balanced Salt Solution (HBSS), and fixed in 4% paraformaldehyde (PAF) at 37 °C for 2 min. The cells were observed using a Leica TCS SP8 confocal system (Leica Microsystems, Wetzlar, Germany) equipped with an HCX PL APO 63X/1.4 NA oil-immersion objective. To simultaneously detect the probes, Br-Sq-C12 was excited with a HeNe laser at 633 nm, whereas Calcein was excited with a DPSS laser at 561 nm. Images were acquired on the three coordinates of the space (XYZ planes) with a resolution of 0.081 μm × 0.081 μm × 0.299 μm and were processed and analyzed with ImageJ software (Rasband, W.S., U.S. National Institutes of Health, Bethesda, MA, USA). Three-dimensional images with Calcein allowed for assessing whether the Br-Sq-C12 encapsulated into QS or in its free form was included within the cellular volume or not.

#### 2.6.3. Statistical Analysis

Data are shown as the average values of three independent pulled experiments ± standard error mean (SEM). Statistical analyses were performed using Graph-Pad Prism 6.0 software (La Jolla, CA, USA). The statistical significance between different conditions was determined by performing a *t*-test or Mann–Whitney test, according to the populations’ distribution (normal or not-normal, respectively). Differences with *p*-values < 0.05 were considered statistically significant and *: *p*-value < 0.05, ***: *p*-value < 0.0005, ****: *p*-value < 0001.

## 3. Results and Discussion

In a previous study, Bromo-Squaraine-C4 (Br-Sq-C4), a non-water soluble indolenine-based dye quaternarized with a four-carbon atom chain, demonstrated successful PDT results in vitro [18,49]. However, Br-Sq-C4 application in biomedicine is hindered by its poor solubility and low chemical stability, especially in aqueous solutions. To overcome this drawback, a promising approach is represented by the incorporation into nanoparticle systems to shield its hydrophobicity, prevent the formation of dye aggregates, and improve solubility in physiological conditions [24,26]. Quatsomes (QS) nanovesicles have shown successful results incorporating different cyanine dyes [42,54,55,56,57], demonstrating long-term stability and biocompatibility. Thus, as a first trial, we prepared Br-Sq-C4-loaded quatsomes (QS). However, the incorporation of Br-Sq-C4 into QS resulted in a limited amount of encapsulated dye (entrapment efficiency was ~50%), as well as significant dye leaking over time (nearly 50% of Br-Sq-C4 was released after one month, see Appendix A). Considering the low stability of this system, Br-Sq-C4 was dismissed, and a new squaraine bearing a longer alkyl chain, i.e., **Br-Sq-C12**, was developed. By the incorporation of the longer alkyl chain, higher hydrophobicity is provided, hypothesizing that this would promote its stable incorporation in the vesicular membranes. This study evidences the importance of the hydrocarbon chain length for the stabilization of the dye in a vesicular membrane, and in particular, into a quatsome nanovesicle.

### 3.1. Synthesis of Br-Sq-C12 and Preparation of Br-Sq-C12-Loaded Quatsomes

The synthesis of Bromo-Squaraine-C12, **Br-Sq-C12** dye (Figure 1 and Figure 1), started with the quaternarization of the bromoindolenine ring (**1**), synthesized following a procedure reported in ref. [49], to obtain compound **2**. This reaction was performed under microwave irradiation and led to increased acidity of the methyl group promoting the following condensation reaction. The final dye was then obtained in a one-step reaction under microwave heating following our well-established method for indolenine-based squaraines [50].

QS composed of Stearalkonium (Stk) and Cholesterol (Chol), with a 1:1 molar ratio Stk/Chol, and loaded with **Br-Sq-C12** were prepared using the DELOS-susp methodology [47] at two different starting concentrations of **Br-Sq-C12**; 200 and 300 μM as initial, pre-processing dye concentration. As a result, two batches of QS encapsulating **Br-Sq-C12** were obtained, named based on their final post-processing dye concentration: QS_Sq_160 and QS_Sq_200, from the 200 and 300 μM, respectively (Figure 2). In addition, non-loaded QS (QS_Blank) was also prepared for comparison with the PS-loaded QS. All formulations were diafiltrated to remove the ethanol and non-entrapped dye or free membrane components from the solution, finally obtaining three batches of water-suspended filtered nanovesicles (see Section 2 for details).

All formulations showed a very similar concentration of membrane components (Table 3). Sample QS_Sq_160 showed a higher average dye encapsulation efficiency (~80%), yielding a final dye concentration of 160 μM, while sample QS_Sq_200 showed, as expected, a higher dye loading in mass (*L*), with an effective dye concentration of ~200 μM at the vesicles.

### 3.2. Physicochemical Properties

QS_Sq_160 and QS_Sq_200 were analyzed with DLS to determine the mean hydrodynamic diameter and polydispersity index (PDI). First, a systematic DLS study was performed in order to elucidate the optimal dilution for QS measurements. As detailed in the section “Systematic DLS study” included at the SI, DLS, and ELS measurements were performed at 1:10 dilution from the final formulation to ensure reliable results. Average hydrodynamic diameter (z-average), PDI, and ζ-potential average values are summarized in Table 3. Both samples showed similar hydrodynamic diameters (~90 nm) and PDI values (<0.2), with a monomodal size distribution (Figure 2b). In addition, plain quatsomes (QS_Blank) display similar characteristics, confirming the high reproducibility of the QS preparation. A highly positive ζ-potential (~70 mV), due to the positive charge of Stearalkonium Chloride, is also comparable among the samples and contributes to the colloidal stability of the nanovesicles [58]. Transmission electron microscopy in cryogenic conditions (cryo-TEM) confirmed the unilamellar vesicle morphology unaffected by the dye encapsulation (Figure 2c–e). Br-Sq-C12-loaded samples showed high homogeneity in the size distribution (Figure 2d,e), as already shown by the low PDI values obtained in DLS. From the analysis of the cryo-TEM images, the average geometric diameter values were estimated as 66 ± 20 nm and 58 ± 18 nm for QS_Sq_160 and QS_Sq_200, respectively (*n* = 100). It is important to keep in mind that the averaged values obtained from cryo-TEM come from representative images, which help to confirm the data obtained from DLS being this last one more statistically representative.

### 3.3. Spectroscopic Characterization

**Br-Sq-C12** shows an absorption maximum at around 640 nm in ethanol (Figure 3a) with a very high molar extinction coefficient (290,000 M^−1^cm^−1^). The UV–Vis spectrum is characterized by a narrow absorption band in the NIR, an essential requirement for the PDT treatment, and a characteristic hypsochromic shoulder typical for polymethine dyes. The main absorption peak is associated with the π→π* HOMO–LUMO transitions, mainly localized on the squaraine core; on the other hand, the shoulder at higher energy can be ascribed to the HOMO-LUMO+1 transition [59,60]. As already observed for other SQs [61], **Br-Sq-C12** shows an excellent fluorescence emission with a maximum emission at 649 nm when dissolved in ethanol, although both the absorption and the fluorescence emission are completely quenched when dissolved in water due to an aggregation caused quenching (ACQ) effect. As shown in Figure 3b,c, the loading into QSs fully overcomes this drawback, increasing the solubility of the dye in aqueous media, with absorbance and fluorescence emission maxima at 644 nm and 655 nm, respectively. The different intensity of the 600 nm shoulder for the free dye in ethanol and the dye-loaded QS is related to the presence of some H-aggregates. However, after the entrapment, the squaraine results are dispersible in 100% water.

The solvatochromic effect on both the dye and dye-loaded QSs absorption and emission spectra were also investigated and reported in Table 4. In general, neither the absorption maxima nor the band shape has been affected by the solvent polarity. A slight difference has been observed in the absorption peak maxima in protic or aprotic solvents; in fact, DMSO induced a 15 nm bathochromic shift in comparison to MeOH, suggesting a higher polarity of the ground state compared to the excited state [62,63,64]. It is worth noticing that both the absorbance and emission maxima of the dye-loaded QSs, irrespective of the amount of incorporated dye, are very close to the values obtained with the free **Br-Sq-C12**, suggesting that the association with the QS did not change the energies and relative probabilities of the electronic transitions. The fluorescence quantum yield of **Br-Sq-C12** in ethanol is in the ranges typical for squaraines in organic media. On the contrary, the QY in water cannot be detected due to the complete insolubility of the dye in aqueous media. However, after the entrapment in the QS, we were able to measure the QY in aqueous dispersion. The values are nevertheless very low, probably due to the presence of some H-aggregates, as also evidenced in the UV–Vis spectra reported in Figure 3b,c. The **Br-Sq-C12** fluorescence lifetime showed a monoexponentially decay and is in the nanoseconds range, as already observed for several squaraines [59].

By comparing the values obtained for QS_Sq_160 and QS_Sq_200 in water, bi-functional equations were necessary to fit the decay curves, suggesting that two different types of interactions occurred with the QSs. Specifically, one fluorescence lifetime is slightly longer, while the other ones (ca. 85%) increased by ca. 2.7 times compared to the free dye. This longer decay could be ascribed to the decrease in rotational/twisting degrees of freedom, confirming a good degree of dye entrapment into the QS vesicles. On the other hand, the shorter lifetime could be due to the presence of a small amount of free dye on the QS surface, leading to a detrimental effect caused by the interaction with highly polar media, such as water [59]. From that data, we can conclude that **Br-Sq-C12** has been successfully incorporated into the QS membrane, allowing its fluorescence in an aqueous media.

### 3.4. Evaluation of Colloidal Stability and Photostability

Previous works on QS have already demonstrated the long colloidal stability, up to a year, of these nanovesicles [41,42,43]. In this work, we have evaluated the stability over time and photostability of the **Br-Sq-C12**-loaded QS with DLS and fluorescence spectroscopy, respectively. The obtained data proved the dye-loaded nanovesicular systems to be very stable for 18 months, with hydrodynamic diameter values around 90 nm and optimally low PDI values maintained around 0.2 (Figure 4a and Appendix A). As previously mentioned, the high positive ζ-potential plays a crucial role in providing colloidal stability; thus, +70 mV ζ-potential values after 10 weeks for both dye-loaded systems demonstrated the long-term stability of those nanovesicles (Figure 4b).

Similarly, photostability was evaluated with periodical UV–vis absorbance measurements for up to 18 months. We noticed a lowering of the main absorbance peak at 644 nm during the time for both samples in the study (Figure 4c,d and Appendix A), indicating a decrement in the PS concentration at the QS nanovesicle. The residual dye concentration encapsulated in QS was quantified again 4 months after the production, and the obtained values were ~130 μM for sample QS_Sq_160 and ~180 μM for sample QS_Sq_200, resulting in a percentual dye leaking of ~20% and ~9% in 4 months, respectively (Figure 4e). In order to better understand this phenomenon, we followed the variation of the peaks’ amplitude over time (ratio peak_660nm_/shoulder_600nm_), which can be indicative of the formation of dye aggregates. The results, presented in Appendix A, show that both bands progressively decrease over time with no significant change in the ratio, suggesting that there is no significant dye aggregation in the QS membrane. Given the obtained results, we can assume that the stability of the dyes is not compromised in a major way by their inclusion in the QS membrane at the obtained loadings.

### 3.5. ROS Production

A preliminary evaluation of the ability of both the free **Br-Sq-C12** and dye-loaded QS to generate Reactive Oxygen Species (ROS) was carried out by using 1,3-diphenylisobenzofuran (DPBF) as a probe [49]. DPBF rapidly reacts with ROS generated by the light-activated dye, resulting in the disappearance of DPBF’s characteristic absorption band at 415 nm due to the formation of the colorless o-dibenzoylbenzene derivative. As a function of the irradiation time, the decrease in the DPBF absorption band at 415 nm has been compared to the values obtained by irradiating a standard, the efficient and well-known ROS generator Rose Bengal (RB). As shown in Figure 5, both the free squaraine and the squaraine loaded into QSs possess faster and higher ROS generation ability compared to the RB. In particular, **Br-Sq-C12** is able to promote the complete decay of DPBF absorption within 180 s, while the same result was obtained in 10 min for reference RB. This fast ROS generation could be ascribed to the presence of bromine, which may facilitate the singlet to triplet state intersystem crossing due to the well-known heavy atom effect [18,19]. More importantly, the entrapment of **Br-Sq-C12** into the QSs does not interfere with the ability of the dye to generate ROS rapidly, an essential requirement for PDT activity.

### 3.6. Cytotoxicity and PDT Assays

Despite the outstanding properties of QS, the remarkable cytotoxicity of quaternary ammonium surfactants, including Stk, could represent a challenge in their in vivo application [65,66]. Therefore, to assess QS biocompatibility, we first performed cell viability assays on MCF-7 cells treated with two different concentrations of blank QS, i.e., 10 and 2 μg/mL (Stk/Chol); see Section 2.6.1 for details. As shown in Figure 6a, QS diluted to a final membrane components’ concentration of 10 μg/mL revealed marked cytotoxicity starting 24 h after treatment and occurring up to 72 h. On the contrary, QS at 2 μg/mL showed good biocompatibility on MCF-7 cells (Figure 6b). QS_Sq_160 and QS_Sq_200 were diluted to 2 μg/mL membrane components, corresponding to a final dye concentration of 68 nM and 85 nM, respectively. Interestingly, the newly synthesized **Br-Sq-C12**-85nM-loaded quatsomes (QS_Sq_200) are slightly more cytotoxic than QS_Blank at 24 and 48 h after treatment with 2 μg/mL, although still highly biocompatible compared to QS_Blank at 10 μg/mL (Figure 6c). This effect is in agreement with previously reported data on polymethine dyes loaded in solid lipid nanoparticles [24]. Consequently, the measurement of the photoactivity of **Br-Sq-C12**-loaded QS was performed by diluting the formulation up to a membrane components’ concentration of 2 μg/mL to avoid any non-targeted cytotoxicity provided by the carrier itself.

The photoactivity of the molecules was quantified by irradiating MCF-7 cells for 15 min in the presence of free **Br-Sq-C12** or **Br-Sq-C12** incorporated into the previously described QS nanosystem (Figure 7). Interestingly, Sq-loaded QS are significantly more phototoxic than free Sq, even at the lowest concentration tested. In fact, **Br-Sq-C12,** in its free form, showed no phototoxicity at any of the tested concentrations (data shown in Appendix A). Of all formulations, QS_Sq_160 (at a final **Br-Sq-C12** concentration of 68 nM) was found to be the most active formulation, as it was significantly different from free **Br-Sq-C12** at all monitored time points (Figure 7). On the contrary, QS_Sq_200 (corresponding to 85 nM effective dye concentration) did not yield a significant improvement in the photoactive properties of the nanosystem after irradiation. Those results point out that a higher concentration of photosensitizer in nanovesicles does not always correlate to higher PDT efficiency. We suspect that **Br-Sq-C12** molecules in the QS_Sq_200 suffer from aggregation due to the higher concentration compared to QS_Sq_160, leading to a decrease of the squaraine photochemical properties and, in consequence, a lower PDT efficiency.

Of note, the concentration of free squaraine usually employed for PDT studies ranges from 1 to 100 μM [49,67,68]. Here, we show that the use of a nanocarrier system allows using much lower concentrations of photosensitizer (up to 10–1000 times lower). In agreement with previous data obtained on other types of nanosystems [24], our results demonstrate that the encapsulation within QS highly enhances the dye’s photoactivity. It is likely that the nanocarrier, by significantly increasing the local concentration of the dye, in addition to reducing its aggregation phenomena, improves its overall spectroscopic properties. Furthermore, taking into account our results on ROS production (Figure 5), it is also possible to hypothesize that the relatively low photoactivity of the free dye in contact with MCF-7 cells may be due to a failure of the molecule to enter the cell, contrary to the nanoparticle system. To test this hypothesis, we performed confocal laser scan microscopy experiments on MCF-7 cells treated O/N with QS_Sq_200 and the corresponding amount of Br-Sq-C12 in its free form (85 nM). More specifically, we labeled the whole cellular volume using Calcein (illustrated in red), and, **Br-Sq-C12** is shown in blue (Figure 8). We acquired the images in three-dimensional space (XYZ), allowing the 3D cellular volume reconstruction and elucidating whether the **Br-Sq-C12** (encapsulated or in its free form) was included within the cellular volume. As shown in Figure 8a, we found that Sq-loaded QS are efficiently internalized by MCF-7 after O/N incubation as demonstrated by the orthogonal view, revealing probe signals included within the Calcein-labeled cell volume (orthogonal view in Figure 8a). By contrast, as previously hypothesized, the Br-Sq-C12 in its free form at the same concentration present in the **Br-Sq-C12**-loaded QS samples was not properly internalized (Figure 8b). This confirms the absence of photoactivity due to hindered cellular uptake of the dye in its free form.

Taken together, these results point out the advantage that nanoencapsulation supposes for photosensitizers such as squaraines. First, the entrapment of Br-Sq-C12 into a nanovesicular system such as QS allows not only its dispersity and stability in aqueous media but also the protection of its photophysical characteristics. Secondly, cell permeability issues that the PS faces in its free form can be overcome by its encapsulation (i.e., QS uptake through phagocytosis [42]). As a result, higher phototoxicity is observed for **Br-Sq-C12**-loaded QS than for the free dye.

## 4. Conclusions

A novel formulation based on Cholesterol–Sterealkonium QS loaded with a squaraine dye (i.e., Br-Sq-C12) has been proposed as an alternative photosensitizer in PDT. QS were prepared with green and scalable technology in a single step; the membrane components are available at pharmaceutical grade and readily adaptable for the incorporation of active targeting ligands. Considering the poor water solubility of squaraine dyes, their loading into QS offers an interesting strategy to bring them into aqueous media, enabling their use for bioapplications. Br-Sq-C12-loaded QS are stable for at least 18 weeks in aqueous media, showing modest dye leaking over time and weak photo instability. Moreover, we have demonstrated that Br-Sq-C12 incorporation into QS does not compromise its photophysical characteristics nor the ability of the dye to generate ROS rapidly, an essential requirement for PDT activity. Indeed, both the free Br-Sq-C12 and Br-Sq-C12-loaded QS possess faster and higher ROS generation ability compared to the well-known ROS generator Rose Bengal. Phototoxicity assays demonstrated that MCF-7 cells internalize Br-Sq-C12-loaded QS, and upon irradiation, an increase in the apoptotic/necrotic cell population is observed. The higher phototoxicity observed for the QS-loaded Br-Sq-C12 vs. its free form can be explained by the higher efficiency in PS delivery. QS provides more photostability in the aqueous media, higher cellular uptake, and significantly increases the local concentration of the PS. Indeed, the use of the nanocarrier system allows using much lower concentrations of photoactive dye (up to 10–1000 times lower). Taken together, our in vitro results support the applicability of QS as nanocarriers for PDT application and highlight the benefits of encapsulating squaraine dyes into stable nanostructures to optimize their photoactive properties.

## Data Availability

Not applicable.

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
