# Peer review of "Quatsomes Loaded with Squaraine Dye as an Effective Photosensitizer for Photodynamic Therapy"

_pharmaceutics, 2023, doi:10.3390/pharmaceutics15030902_

Round 1
Reviewer 1 Report (Previous Reviewer 3)
Background of this work is explained. This referee recommends publication of this work in Pharmaceutics.
What is Sq_300 in Figure 6d?
Author Response
Reviewer 1
What is Sq_300 in Figure 6d?
We thank the reviewer for pointing out the discrepancy reported in Figure 6. In the new version of the manuscript, this is Figure 7 (Figure 6d in the previous version).
Sq_300 is indeed the Br-Sq-12 concentration in its free form which is the control QS prepared at the highest loading (named QS_Sq_200). As reported in the text (page 14 line 523-529), QS_Sq_160 and QS_Sq_200 were diluted to 2 μg/mL membrane components, corresponding to a final dye concentration of 68 nM and 85 nM of free dye respectively.
We now have renamed the free Sq control used in Fig 7 (previous Fig. 6d) to be more consistent and clearer:
- Sq_200 = Free form of the squarine at the same concentration of the QS_Sq_200
- Sq_160 = Free form of the squarine at the same concentration of the QS_Sq_160
We have also corrected the supplementary Fig S6 accordingly.

Reviewer 2 Report (New Reviewer)
In the present manuscript, Quatsomes loaded with squaraine dye as an effective photosensitizer for Photodynamic Therapy, by Bordignon et al., the authors have used a set of experiments were made to characterize the dye’s encapsulation in the nanocontainer and assess its ability for PDT, in vitro.
Using non-toxic, thermodynamically stable nanometric quatsomes as nanocontainers to help surpass water solubility/aggregation problems with low dye concentration is a useful strategy. Overall, the data seem sound and are presented in a readable manner.
I find this work of potential interest to readers of Pharmaceutics. Nonetheless, some points need to be addressed before publication:
1) I find the Abstract too generic. It could be improved by focusing more on the work and data obtained.
2) In the Introduction, page 3, the information in lines 98 to 111 is repeated in the Results and Discussion section on page 8 (lines 344 to 361). In my opinion, I would mention it only in the Introduction section.
3) On page 11, line 429, the authors mention the “intensity of the 660 nm shoulder…”, but I believe they mean “the 600 nm shoulder…”.
4) Are the quatsomes obtained using each solvent mentioned (ethanol, DMSO, acetone...) as the dispersant medium? Are the structural vesicular characteristics of quatsomes like those observed in water?
5) The authors should clarify how they conclude that the “longer decay could be ascribed to the decrease in rotational/twisting degrees of freedom…”. It is known that the non-radiative processes of squaraines are essentially controlled by polarity.
6) “Bengal Rose” should be changed to “Rose Bengal” (e.g.: page 13, line 498, etc).
7) In figure 7, the meaning of a) QS_Sq and b) Sq) is missing.
8) References [5] and [6] are repeated.
Author Response
Reviewer 2
In the present manuscript, Quatsomes loaded with squaraine dye as an effective photosensitizer for Photodynamic Therapy, by Bordignon et al., the authors have used a set of experiments were made to characterize the dye’s encapsulation in the nanocontainer and assess its ability for PDT, in vitro.
Using non-toxic, thermodynamically stable nanometric quatsomes as nanocontainers to help surpass water solubility/aggregation problems with low dye concentration is a useful strategy. Overall, the data seem sound and are presented in a readable manner.
I find this work of potential interest to readers of Pharmaceutics. Nonetheless, some points need to be addressed before publication:
1) I find the Abstract too generic. It could be improved by focusing more on the work and data obtained.
We have revised the abstract, as well as Pharmaceutics’ guidelines for the abstract contents. As proposed, we have followed the structure: 1) Background, 2) Methods, 3) Results, and 4) Conclusion. Since the abstract is already close to the maximum length, we have slightly modified it presenting another relevant result regarding ROS production (underlined).
Photodynamic therapy is a non-invasive therapeutic strategy that combines external light with a photosensitizer (PS) to destroy abnormal cells. Despite the great progress in the development of new photosensitizers with improved efficacy, the PS’s photosensitivity, high hydrophobicity, and tumor target avidity still represent the main challenges. Herein, newly synthesized brominated squaraine – exhibiting intense absorption in the red/near-infrared region – has been successfully incorporated into Quatsome (QS) nanovesicles at different loadings. The formulations under study have been characterized and interrogated in vitro for cytotoxicity, cellular uptake, and PDT efficiency in a breast cancer cell line. The nanoencapsulation of brominated squaraine into QS overcomes the non-water solubility limitation of the brominated squaraine without compromising its ability to rapidly generate ROS. In addition, PDT effectiveness is maximized due to the highly localized PS loadings in the QS. This strategy allows using a therapeutic squaraine concentration that is 100 times lower than the concentration of free squaraine usually employed in PDT. Taken together, our results reveal the benefits of the incorporation of brominated squaraine into QS to optimize their photoactive properties and support their applicability as photosensitizer agents for PDT.
2) In the Introduction, page 3, the information in lines 98 to 111 is repeated in the Results and Discussion section on page 8 (lines 344 to 361). In my opinion, I would mention it only in the Introduction section.
We thank the reviewer for his/her comment. However, we added this explanation following a previous reviewer’s suggestion. We leave the decision to the editor.
3) On page 11, line 429, the authors mention the “intensity of the 660 nm shoulder…”, but I believe they mean “the 600 nm shoulder…”.
The reviewer is right, we have changed the value. The shoulder wavelength is 600 nm (line 429).
4) Are the quatsomes obtained using each solvent mentioned (ethanol, DMSO, acetone...) as the dispersant medium? Are the structural vesicular characteristics of quatsomes like those observed in water?
Quatsomes are obtained using the DELOS-susp method which involves an organic phase (ethanolic solution) depressurized over an aqueous phase (Milli-Q water). During the preparation of QS, no other solvent than ethanol and water is employed.
Other organic solvents such as DMSO or acetone are employed during the synthesis of the Br-Sq-C12, after solvent evaporation Br-Sq-C12 is solubilized in ethanol and added to the solution containing all membrane components. To make this point clearer we have revised section 2.2 Preparation of dye-loaded Chol/Stk QS by DELOS-susp.
The structural vesicular characteristics of quatsomes are described in water since it is the media in which nanovesicles are produced.
5) The authors should clarify how they conclude that the “longer decay could be ascribed to the decrease in rotational/twisting degrees of freedom…”. It is known that the non-radiative processes of squaraines are essentially controlled by polarity.
The reviewer is right. LT and QY are also dependent on solvent polarity, but as a general trend for SQ dyes, LT and QY usually decrease with increasing solvent polarity. However, in this specific case, we can not refer to this statement, due to the impossibility to acquire and compare the LT in water (the molecule is not soluble in water). In this specific case, the comparison is among the LT of the free dye in Ethanol and the LT of the dye incorporated into QS and acquired in water. Therefore, our conclusions were driven by what we found in the literature for SQ dyes incorporated in other nanosystems. To better clarify, we added the references related to our conclusions (line 465).
6) “Bengal Rose” should be changed to “Rose Bengal” (e.g.: page 13, line 498, etc).
We thank the reviewer for the comment. We changed the name “Rose Bengal” within all the manuscript as well as in Figure 5.
7) In figure 7, the meaning of a) QS_Sq and b) Sq) is missing.
We thank the reviewer for the comment and apologize for the missing information. We have added the explanation of the QS_Sq and Sq in the Figure 8 caption (previous Figure 7). Indeed the confocal images were acquired using the QS_Sq_200 formulation and its corresponding free Br-Sq-C12 concentration (85nM).
8) References [5] and [6] are repeated.
We appreciate the reviewer’s comments and the references section has been carefully revised and corrected accordingly.

Reviewer 3 Report (New Reviewer)
The manuscript ID: pharmaceutics-2232887 “Quatsomes loaded with squaraine dye as an effective photo-sensitizer for Photodynamic Therapy.”. In this article, the author employed non-liposomal 85 nanovesicles, named quatsomes (QS), as a nanocarrier for a hydrophobic squaraine dye. Br-Sq-C12-loaded QS are stable for at least 18 weeks in aqueous media, showing modest dye leaking over time and weak photo instability and they demonstrated that Br-Sq-C12 incorporation into QS does not compromise its photophysical characteristics, nor the ability of the dye to rapidly generate ROS, an essential requirement for the PDT activity. I think that their presentation and explanation of paper writing were not so exciting. But the topic is interesting for the referee. Therefore, I recommend publication only after minor revisions.
1) The introduction is not impressive in this paper, the author needs to include more introduction to PDT and explains. I think that the author needs to include the PDT system and working principle diagram in the introduction part for better understanding. (See lines 43 to 48)
2) What are the best suitable photosensitizer used in PDT?
3) Page No.2; Line 67-70: the author has written “organic dyes, such as polymethine dyes (squaraine and cyanine dyes) represent a promising alternative to PS thanks to the higher selectivity, purity, and absorption at longer wavelengths compared to porphyrin-derived PS”.
4) My question is most organic dyes are not dissolving in water. So, only a few sensitizers will dissolve in water. What they are? Include in the manuscript explanations.
5) Figure 1b image is not good, the author needs to include a high-resolution image.
6) In scheme 1; the author needs to include Presser (P) for MW.
7) The author has written, “The compound solubility proved too low to record a 13C NMR spectrum”. Interesting and why the author has not tried other solvents of recording 13C NMR. Many SQ sanitizers have reported 13C nmrs with various solvents. I think that better to include the 13C nmr by using other solvents.
8) the author needs to include IR spectra for better understanding.
9) I did not find a lifetime figure. The author needs to include lifetime graphs and explains what are the changes found.
Author Response
Reviewer 3
The manuscript ID: pharmaceutics-2232887 “Quatsomes loaded with squaraine dye as an effective photo-sensitizer for Photodynamic Therapy.”. In this article, the author employed non-liposomal 85 nanovesicles, named quatsomes (QS), as a nanocarrier for a hydrophobic squaraine dye. Br-Sq-C12-loaded QS are stable for at least 18 weeks in aqueous media, showing modest dye leaking over time and weak photo instability and they demonstrated that Br-Sq-C12 incorporation into QS does not compromise its photophysical characteristics, nor the ability of the dye to rapidly generate ROS, an essential requirement for the PDT activity. I think that their presentation and explanation of paper writing were not so exciting. But the topic is interesting for the referee. Therefore, I recommend publication only after minor revisions.
1) The introduction is not impressive in this paper, the author needs to include more introduction to PDT and explains. I think that the author needs to include the PDT system and working principle diagram in the introduction part for better understanding. (See lines 43 to 48)
We thank the reviewer for his/her comments. In the introduction, in addition to explaining the basics in the first paragraph (lines 43 - 56), we have cited some references related to the working principle of the PDT. Considering that PDT has been described for more than 60 years in the scientific literature, there are a great number of reviews explaining the working principle behind PDT. We believe that, in this context, it is not useful to extend the introduction section with information that can be easily found in these reviews.
In this regard, we have added one reference that will help the readers (Basic principles of photodynamic therapy, by Ian J. MacDonald and Thomas J. Dougherty) at line 44. In addition, we have modified Figure 1 including a basic scheme of the PDT working principle (Figure 1c).
2) What are the best suitable photosensitizer used in PDT?
Unfortunately, it is very hard to answer this question, since an ideal PSs does not exist. Moreover, making comparisons among the results reported in the literature for different PSs is quite difficult: the treatment conditions are not standardized and the treatment is often personalized.
3) Page No.2; Line 67-70: the author has written “organic dyes, such as polymethine dyes (squaraine and cyanine dyes) represent a promising alternative to PS thanks to the higher selectivity, purity, and absorption at longer wavelengths compared to porphyrin-derived PS”. My question is most organic dyes are not dissolving in water. So, only a few sensitizers will dissolve in water. What they are? Include in the manuscript explanations.
This topic has been widely reviewed in many reviews reported in the literature and cited in the manuscript. We did not mention them to avoid a long and redundant discussion. Moreover, some of the water-soluble PSs reported in the literature are dissolved in water with a certain percentage of DMSO. Another option is the insertion of functional groups that can increase water solubility, but in this case, the photochemical properties get worse. To obtain a compromise, a good option is their incorporation into different NPs. This discussion has been included in the manuscript (line 78-80):
“[...] soluble PS in either the hydrophobic core of micelles [28] or the membrane of liposomes [29,30]. It is worth noting that other strategies such as the insertion of functional groups into the PS structure to increase water solubility have been reported [31]. However, those modifications usually compromise the photochemical properties of the PS [32]. Thus, nanoencapsulation of the PS improves its solubility in the aqueous environment without altering its chemical structure and, benefits from either active targeting, with the functionalization of targeting [...]”
5) Figure 1b image is not good, the author needs to include a high-resolution image.
Figure 1 has been changed in line with the first comment from this same reviewer, including also a higher-resolution image.
6) In scheme 1; the author needs to include Presser (P) for MW.
We did not completely understand the reviewer’s request. We checked on the manual instrument but we were not able to find the “presser”.
7) The author has written, “The compound solubility proved too low to record a 13C NMR spectrum”. Interesting and why the author has not tried other solvents of recording 13C NMR. Many SQ sanitizers have reported 13C nmrs with various solvents. I think that better to include the 13C nmr by using other solvents.
We thank the reviewer for the suggestion. We tried to solubilize the molecule in different deuterated solvents (DMSO, CDCl3, MeOD, Acetone-d6), but none of them provided the proper solubility to acquire a 13C NMR spectrum.
8) the author needs to include IR spectra for better understanding.
We thank the reviewer for the suggestion. However, for the purpose of the present work, we did not add the FTIR spectra, since we reported the NMR and mass spectrometry spectra (Figures S2 and S3, respectively) to confirm the purity of the synthesized molecules.
9) I did not find a lifetime figure. The author needs to include lifetime graphs and explains what are the changes found.
Usually, lifetime figures are not included in the paper, since they did not provide additional information. Lifetime measurements are reported as numbers in tables, as we did in Table 4. The changes among the different LTs are explained in the main text.
We have attached a file with the lifetime decay graphs for the revision purpose, not for publication (see the following page).

This manuscript is a resubmission of an earlier submission. The following is a list of the peer review reports and author responses from that submission.
Round 1
Reviewer 1 Report
The manuscript titled "quatsomes loaded with squaraine dye as an effective photosensitizer for photodynamic therapy" presents the study of a novel formulation based on cholesterol-sterealkonium QS loaded with a squaraine dye as an alternative photosensitizer in PDT. Well designed experiments with careful analysis support their propose that the Br-Sq-C12 loaded QS have a great potential for application and further development. However, few revisions are required prior to publication:
1. The NMR spectrum and Mass spectrum should be provided for compound Br-Sq-C12 to prove the successful synthesis of this compound and to prove the high purity of the PS used for quatsome study.
2. As the author mentioned "Sq-loaded QS are significantly more phototoxic than free Sq" and "the relatively low photoactivity of the free dye in contact with MCF-7 cells may be due to failure of the molecule to enter cell", I suggest the author to provide cell imaging of MCF-7 cells incubated with free Sq and QS loaded with Br-Sq-C12 to test this hypothesis. It is also important to show that cells are able to uptake these quatsomes.
3. Line 540, change "15'" to "15 min" to avoid confusion.
4. The author should provide much more reference for the introduction part to support their statement and for readers to follow.
5. Language and gramma isssues should be carefully checked, for example, line 60 should be "most of them are porphyrin-derived".
Author Response
Journal: Pharmaceutics (ISSN 1999-4923)
Manuscript ID: pharmaceutics-2055333
Type: Article
Title: Quatsomes loaded with squaraine dye as an effective photo-sensitizer for Photodynamic Therapy
Authors: Nicolò Bordignon, Mariana Köber, Giorgia Chinigò, Carlotta Pontremoli, Ettore Sansone, Guillem Vargas-Nadal, Maria Jesus Moran Plata, Alessandra Fiorio Pla, Nadia Barbero* , Judit Morla-Folch* , Nora Ventosa *
Section: Nanomedicine and Nanotechnology
Special Issue: Fluorescent Organic Nanoparticles for Bioimaging and Theragnostics
Reviewer 1:
The manuscript titled "quatsomes loaded with squaraine dye as an effective photosensitizer for photodynamic therapy" presents the study of a novel formulation based on cholesterol-sterealkonium QS loaded with a squaraine dye as an alternative photosensitizer in PDT. Well designed experiments with careful analysis support their propose that the Br-Sq-C12 loaded QS have a great potential for application and further development. However, few revisions are required prior to publication:
- The NMR spectrum and Mass spectrum should be provided for compound Br-Sq-C12 to prove the successful synthesis of this compound and to prove the high purity of the PS used for quatsome study.
The reviewer is right, NMR and mass spectrum have been provided in Figure S2 and S3, respectively. The reference to the Figures has been also added to the text in Section 2.1.2.
- As the author mentioned "Sq-loaded QS are significantly more phototoxic than free Sq" and "the relatively low photoactivity of the free dye in contact with MCF-7 cells may be due to failure of the molecule to enter cell", I suggest the author to provide cell imaging of MCF-7 cells incubated with free Sq and QS loaded with Br-Sq-C12 to test this hypothesis. It is also important to show that cells are able to uptake these quatsomes.
We agree with the reviewer on both observations. Therefore, we have tested the hypothesis by performing confocal laser scan microscopy experiments (for details see Material & Methods section 2.6.2) on MCF-7 treated with both free Sq-C12 and QS loaded with Br-Sq-C12. Cell internalization of the QS loaded with Br-Sq-C12 is shown in Figure 7 as well as discussed in section 3.6 “Cytotoxicity and PDT assays”. We found that Sq-loaded QS are efficiently internalized by MCF-7 after O/N incubation.
This outcome support previous results where we showed the cell uptake of quatsomes on HeLa and CHO-K1 cells (ACS Appl. Mater. Interfaces 2020, 12, 18, 20253–20262) as well as Saos2 cells (Chem. Eur. J. 2018, 24, 11386).
- Line 540, change "15'" to "15 min" to avoid confusion.
Proposed change it’s been modified in the text.
- The author should provide much more reference for the introduction part to support their statement and for readers to follow.
In agreement with the reviewer’s comment, we have gone through the introduction and added more references relevant to the study and modified the text accordingly. The references and text reviewed are mainly related to the current PDT challenges and strategies to overcome them, such as the use of new photosensitizers with higher performance or their encapsulation.
This better supports our statement and help the reader to follow the importance of not only developing new photosensitizers but also the advantages of their nanoencapsulation.
To see the changes please check the manuscript version with marked changes.
- Language and gramma issues should be carefully checked, for example, line 60 should be "most of them are porphyrin-derived".
We checked carefully the manuscript and corrected language and grammar.

Reviewer 2 Report
The abstract should present the study findings and conclusions.
the introduction is lacking a strong rationale support by current evidence, arguments lack the support of references.
the methdology is too long, provide references for the stand methods used from previous studies. simplify and concise the methdology. the company name and commenrcial products used in the methods must be mentioned in paranthesis.The sample sizes are not mentioned in the different experimentions.
All tables shoudl mention the units of the numbers mentioned in it.
as section 3 is results, so discussion appears to be missing completely. are results and discussion presented combined ? not clear.
Add discussion with reasosn for the findings and comparison to other research findings. also provide evidence to support your outcomes. the hypothesis was defined ? not clear, was it accepted ?
the use clinically , how will that be possible. what are the recomendation. simplify the conclusions and make them factual. future studies and clinical recomendation shoudl be in discussion section. the images are poor quality , the reviewer cannot evalute the results as figures, 2,3 ,4 and 6 are not readable at all.
Author Response
Journal: Pharmaceutics (ISSN 1999-4923)
Manuscript ID: pharmaceutics-2055333
Type: Article
Title: Quatsomes loaded with squaraine dye as an effective photo-sensitizer for Photodynamic Therapy
Authors: Nicolò Bordignon, Mariana Köber, Giorgia Chinigò, Carlotta Pontremoli, Ettore Sansone, Guillem Vargas-Nadal, Maria Jesus Moran Plata, Alessandra Fiorio Pla, Nadia Barbero* , Judit Morla-Folch* , Nora Ventosa *
Section: Nanomedicine and Nanotechnology
Special Issue: Fluorescent Organic Nanoparticles for Bioimaging and Theragnostics
Reviewer 2:
The abstract should present the study findings and conclusions.
In agreement with the reviewer comment we have updated the abstract, marked in bold as follows:
[…]. “QSs loaded with squaraine showed long-term stability and high entrapment efficiency, providing high local concentrations of the PS at the nanoscale and in consequence, improved therapeutic efficacy. Moreover, the nanoencapsulation of squarine into QS allowed using squarine concentrations that are 100 time lower than the concentration of free squarine usually employed in PDT. Taken together, our results support the applicability of QS as nanocarriers for PDT […] ”
The introduction is lacking a strong rationale support by current evidence, arguments lack the support of references.
In agreement with the reviewer’s comment, we have gone through the introduction and added more references relevant to the study as well as modified the text. The references and modified text are mainly related to the current PDT challenges and strategies to overcome them, such as the use of new photosensitizers with higher performance or their encapsulation.
This better supports our statement and help the reader to follow the importance of not only developing new photosensitizers but also the advantages of their nanoencapsulation.
To see the changes please check the manuscript version with marked changes.
The methdology is too long, provide references for the stand methods used from previous studies. simplify and concise the methdology. The company name and commercial products used in the methods must be mentioned in paranthesis.
The methodology is reporting the synthesis of the organic compounds and of the QS as well as the dye-loading procedures. All the physicochemical and spectroscopic characterizations together with the biological assay protocols are reported in this part.
In order to shorten it, part of the methodology has been moved to the SI (mainly the section 2.2 Preparation of dye-loaded Chol/Stk QS by DELOS-susp). Other section such as 2.3.1 and 2.4.1 have been rewritten in order to summarize it.
Company names and commercial products have been included.
The sample sizes are not mentioned in the different experimentions.
All tables should mention the units of the numbers mentioned in it.
We appreciate the comments from the reviewer, sample sizes have been added in the table descriptions. Moreover, all tables have been revised and all the units are reported except for dimensionless numbers.
as section 3 is results, so discussion appears to be missing completely. are results and discussion presented combined? not clear.
The referee is right, we have corrected section 3 as “Results and Discussion”. In addition, we have added a part in the discussion where results from cellular internalization studies are discussed (lines 957 - 971).
Add discussion with reasons for the findings and comparison to other research findings. also provide evidence to support your outcomes. the hypothesis was defined? not clear, was it accepted?
Tackling that point, we have reviewed different sections of the manuscript in order to make our hypothesis and findings clearer for the reader.
First, we have done some changes in the introduction (line 99-101).
“By the incorporation of the C12 hydrocarbon chain, Br-Sq can be anchored to the QS membrane more stably due to its larger lipophilicity in comparison with Br-Sq-C4. In consequence, the loading of the squaraine into the QS nanovesicles can be improved as well as the PDT effectiveness. Herein, we present the design of new QS composed […]”
Secondly, in section “3.4 Evaluation of colloidal stability and photostability” we have added Figure S4 to demonstrate the long-term stability (18 months) of the quastomes, as hypothesized in the introduction.
Moreover, we have also included internalization results in order to prove our hypothesis of quatsomes cellular uptake as mentioned in the text (line 543 - 550). We have tested the hypothesis by performing confocal laser scan microscopy experiments (for details see Material & Methods section 2.6.2) on MCF-7 treated with both free Sq-C12 and QS loaded with Br-Sq-C12. Cell internalization results are shown in Figure 7 as well as discussed in section 3.6 “Cytotoxicity and PDT assays”, with the text as follows (line 547 - 561):
“To test this hypothesis, we performed confocal laser scanning microscopy experiments on MCF-7 cells treated O/N with QS_Sq_200 and the corresponding amount of Br-Sq-C12 in its free form (85 nM). More specifically, we labeled the whole cellular volume using Calcein (red signals in Figure 7) and we acquired the images on the three coordinates of the space (XYZ), thus being able to reconstruct the 3D cellular volume and therefore to check whether the Sq encapsulated or in its free form (blue signals in Figure 7) was included within the cellular volume or not. As shown in Figure 7a, we found that Sq-loaded QS are efficiently internalized by MCF-7 after O/N incubation as demonstrated by the orthogonal view revealing probe signals included within the calcein-labelled cell volume (orthogonal view in Figure 8a). By contrast, as previously hypothesized, the Br-Sq-C12 in its free form at the same concentration present in the QS-Sq samples was not properly internalized by MCF-7 cells (Figure 7b), thus confirming that the absence of photoactivity observed on MCF-7 is mainly due to hindered cellular uptake of the dye in its free form.”
The use clinically, how will that be possible. what are the recomendation. simplify the conclusions and make them factual. future studies and clinical recomendation shoudl be in discussion section.
We thank the reviewer for the comment. The aim of the present work is to clarify the potential of squaraines in the photodynamic therapy and the applicability of QS as nanocarriers for PDT. The use of QS allows to achieve a high SQ concentration in the pathological site and optimize its photoactive properties. In fact, as highlight in the introduction, one of the main issues related to this class of dyes is their poor chemical stability and solubility in biological environment, which still limits the spreading of their application in PDT. Since QSs have been reported to be safe and non-toxic for biomedical applications and remain stable for years, the authors believe that the incorporation of SQ into QSs represents a promising strategy to overcome stability/solubility problems, proposing an innovative nanosystem with potential application in PDT.
Here we reported some evidence that confirm our hypothesis, however, more efforts need to be devoted in the field of PMDs, in particular in vivo studies and in pre-clinical trials need to be evaluated, to clarify their real potential in the photodynamic therapy.
To better clarify this point, the following sentences have been added in the conclusions section:
“[…] Moreover, the proposed nanosystem will allow to modify or functionalize NPs surfaces with the idea to exploit external stimuli such as the pH and hypoxic tumor microenvironments, thus obtaining powerful tools to enrich the current stock of the permanently used systems. It is worth highlighting that despite the evidence that polymethine dyes represent a promising alternative as photosensitizers for PDT, further efforts are required to optimize the most suitable concentration to meet the proper biocompatibility and to evaluate the in vivo efficacy with a view to future biomedical applicability”.
The images are poor quality. The reviewer cannot evalute the results as figures, 2,3 ,4 and 6 are not readable at all.
We apologize for the problem: we replaced the figures with high resolution ones.

Reviewer 3 Report
This paper describes synthesis, physical property, and cell viability of Br-Sq-C12 loaded QS. Previous study on Br-Sq-C4 is extended here to Br-Sq-C12. Unfortunately, no comparative discussion is provided between Br-Sq-C4/QS and Br-Sq-C12/QS, and the work does not tell whether they are improved or not. Without such discussions, this paper cannot be recommended for publication in Pharmaceutics. Several additional comments follow.
1) Figure 6c is not easy to understand.
2) Figure 7. How cell viability is determined, and what does it means the increase of the number in the vertical axis?
Author Response
Journal: Pharmaceutics (ISSN 1999-4923)
Manuscript ID: pharmaceutics-2055333
Type: Article
Title: Quatsomes loaded with squaraine dye as an effective photo-sensitizer for Photodynamic Therapy
Authors: Nicolò Bordignon, Mariana Köber, Giorgia Chinigò, Carlotta Pontremoli, Ettore Sansone, Guillem Vargas-Nadal, Maria Jesus Moran Plata, Alessandra Fiorio Pla, Nadia Barbero* , Judit Morla-Folch* , Nora Ventosa *
Section: Nanomedicine and Nanotechnology
Special Issue: Fluorescent Organic Nanoparticles for Bioimaging and Theragnostics
Reviewer 3:
This paper describes synthesis, physical property, and cell viability of Br-Sq-C12 loaded QS. Previous study on Br-Sq-C4 is extended here to Br-Sq-C12. Unfortunately, no comparative discussion is provided between Br-Sq-C4/QS and Br-Sq-C12/QS, and the work does not tell whether they are improved or not. Without such discussions, this paper cannot be recommended for publication in Pharmaceutics. Several additional comments follow.
We respect the point of view of the reviewer, but we disagree. The purpose of this manuscript is not a comparative study between two dyes but it the characterization of a new photosensitizer-loaded nanovesicle. Moreover, as reported in Section 3 and in Figure S1, the incorporation of Br-Sq-C4 inside QS resulted in a very low entrapment efficiency and a high dye leaking in a short time. Therefore, it was not possible to use these QS loaded with Br-Sq-C4. This is explained in the last section of the introduction, which has been rephrased for clarity:
“As a first trial, non-water soluble Bromo-Squaraine-C4 (Br-Sq-C4), a well-performing squaraine in PDT which has already demonstrated successful results in vitro [40], was incorporated into QSs nanovesicles to enhance its stability. However, Br-Sq-C4 was not stably anchored into the QS membrane and showed significant dye leaking over time, with only 15% of the initial dye concentration remaining after 6 weeks (see Figure S1 in the Supporting Information). In addition, in vitro tests showed no effect on cell viability upon irradiation. Therefore, we designed and synthesized a similar squaraine bearing longer alkyl chains, i.e. Br-Sq-C12 (see Scheme 1). By the incorporation of the C12 hydrocarbon chain, Br-Sq can be anchored to the QS membrane more stably due to its larger lipophilicity in comparison with Br-Sq-C4.”
1) Figure 6c is not easy to understand.
We have changed Figure 6c by representing viability data as in Figure 6a and b, willing to show more understandable results and easier to interpret.
2) Figure 7. How cell viability is determined, and what does it means the increase of the number in the vertical axis?
In Figure 7 (Fig 6d in the revised version), as well as in Figure 6a-c and S3, absorbance values are normalized on the control at 24h, i.e. on the number of viable cells 24h after the treatments, and represented as mean ± SEM of all the replicates assessed. Therefore, the increase of the number in the vertical axis corresponds to the increase of the absorbance at 490 nm = the number of viable cells. The Figure 6 caption has been updated to include such description.
“Figure 6 – Cell viability assays based on a colorimetric method (MTT assay, Abs490nm) on MCF-7. Cells were treated […]”
Regarding how cell viability is determined, we used CellTiter 96® AQueous Non-Radioactive cell proliferation assay (Promega, USA) accordingly to the manufacturer’s instructions, as reported in Material and Methods section 2.6.1. CellTiter 96® AQueous Non-Radioactive cell proliferation assay is a colorimetric method widely used in cell viability/proliferation assays: it is based on the use of a tetrazolium compound (MTS) which, in metabolically active cells, is bioreduced into a colored formazan product (Abs = 490 nm). Consequently, absorbance values recorded are considered as being proportional to the number of viable cells. Briefly, 2h after MTS incubation at 37°C, absorbance at 490 nm was recorded using a microplate reader (FilterMax F5, Multi-Mode Microplate Reader, Molecular Devices). Absorbance values were normalized on the control at 24h and analyzed as being proportional to the number of viable cells.

Round 2
Reviewer 2 Report
The rationale is weak
The topic lacks novelty.
Some of the methods are repeated from previous studies, kindly provide references of the studies. Mention hypothesis. What are the clinicial recomendations based on the outcomes.
The major limitations must be highlighted.
Kindly update references to the last 5-10 years.
The discussion is mostly related to other studies. the results of the surrent studies shoudl be discussed.
Reviewer 3 Report
1) Authors previously reported Br-Sq-C4, and describe here Br-Sq-C12. Authors comparatively note
As a first trial, non-water soluble Bromo-Squaraine-C4 (Br-Sq-C4), a well-performing squaraine in PDT which has already demonstrated successful results in vitro [40], was incorporated into QSs nanovesicles to enhance its stability. However, Br-Sq-C4 was not stably anchored into the QS membrane and showed significant dye leaking over time, with only 15% of the initial dye concentration remaining after 6 weeks (see Figure S1 in the Supporting Information). In addition, in vitro tests showed no effect on cell viability upon irradiation. Therefore, we designed and synthesized a similar squaraine bearing longer alkyl chains, i.e. Br-Sq-C12 (see Scheme 1). By the incorporation of the C12 hydrocarbon chain, Br-Sq can be anchored to the QS membrane more stably due to its larger lipophilicity in comparison with Br-Sq-C4. In consequence, the loading of the squaraine into the QS nanovesicles can be improved as well as the PDT effectiveness.
Unfortunately, they do not compare Br-Sq-C4 and Br-Sq-C12 in main text, and no comparative data is shown in Figure S1. Comparison is a basis of science. Is there any reason that they do not want to describe their improvement?
2) Figure 6c. Does this mean QS Blank (2 microgram/mL) shows highest cytotoxicity? Please explain CTRL, QS_Blank (10 and 2 microgram/mL), Sq 200, and Sq 300. Are they different samples from QS_Sq 160 and 200?
3) This referee cannot well understand experiments at line 279, line 290, and Figure 6 caption. Normalized against what? May be should normalize against blank but not 24 h experiment.
Absorbance values were normalized on the control at 24h and analyzed as being proportional to the number of viable cells.
The photodynamic effect of Br-Sq-C12-loaded QS was evaluated by comparing the viability of cells treated with QS_Sq or with the same concentration of Br-Sq-C12 in its free form upon irradiation.
(Br-Sq-12 samples on irradiated cells not treated; QS_Sq samples on irradiated cells treated with blank QS)
This referee cannot recommend publication of this work in Pharmaceutics in its present form.